# Evaluation of Green Silver Nanoparticles Fabricated by *Spirulina platensis* Phycocyanin as Anticancer and Antimicrobial Agents

**DOI:** 10.3390/life12101493

**Published:** 2022-09-26

**Authors:** Abel-Fattah Salah Soror, Mai Waled Ahmed, Abdalla E. A. Hassan, Mona Alharbi, Nouf H. Alsubhi, Diana A. Al-Quwaie, Ghadeer I. Alrefaei, Najat Binothman, Majidah Aljadani, Safa H. Qahl, Fatima A. Jaber, Hanan Abdalla

**Affiliations:** 1Botany and Microbiology Department, Faculty of Science, Zagazig University, Zagazig 44519, Egypt; 2Chemistry Department, Faculty of Science, Zagazig University, Zagazig 44519, Egypt; 3Department of Biochemistry, College of Science, King Saud University, Riyadh 11451, Saudi Arabia; 4Biological Sciences Department, College of Science & Arts, King Abdulaziz University, Rabigh 21911, Saudi Arabia; 5Department of Biology, College of Science, University of Jeddah, P.O. Box 80327, Jeddah 21589, Saudi Arabia; 6Department of Chemistry, College of Sciences & Arts, King Abdulaziz University, Rabigh 21911, Saudi Arabia

**Keywords:** SPAgNPs, *Spirulina*, phycocyanin, cancer, cell lines, antimicrobial, anticancer

## Abstract

Green nanotechnology has attracted attention worldwide, especially in treating cancer and drug-resistant section 6 microbes. This work aims to investigate the anticancer activity of green silver nanoparticles synthesized by *Spirulina platensis* phycocyanin (SPAgNPs) on two cancer cell lines: Lung cancer cell line (A-549) and breast cancer cell line (MCF-7), compared to the normal human lung cell line (A138). We also aimed to investigate the bactericidal activity against *Staphylococcus aureus* ATCC29737, *Bacillus cereus* ATCC11778, *Escherichia coli* ATCC8379, and *Klebsiella pneumonia*, as well as the fungicidal activity against *Candida albicans* (ATCC6019) and *Aspergillus niger*. The obtained SPAgNPs were spherical and crystalline with a size of 30 nm and a net charge of −26.32 mV. Furthermore, they were surrounded by active groups responsible for stability. The SPAgNPs scavenged 85% of the DPPH radical with a relative increase of approximately 30% over the extract. The proliferation of cancer cells using the MTT assay clarified that both cancer cells (A-549 and MCF-7) are regularly inhibited as they grow on different concentrations of SPAgNPs. The maximum inhibitory effect of SPAgNPs (50 ppm) reached 90.99 and 89.51% against A-549 and MCF7, respectively. Regarding antimicrobial activity, no inhibition zones occurred in bacterial or fungal strains at low concentrations of SPAgNPs and the aqueous *Spirulina platensis* extract. However, at high concentrations, inhibition zones, especially SPAgNPs, were more potent for all tested microorganisms than their positive controls, with particular reference to *Staphylococcus aureus,* since the inhibition zones were 3.2, 3.8, and 4.3 mm, and *Bacillus cereus* was 2.37 mm when compared to tetracycline (2.33 mm). SPAgNPs have more potent antifungal activity, especially against *Aspergillus niger*, compared to their positive controls. We concluded that SPAgNPs are powerful agents against oxidative stress and microbial infection.

## 1. Introduction

Nanotechnology is considered a new branch of science, creating growing excitement in the life sciences, especially in modern medicine [1]. Silver and silver nanoparticles (AgNPs) are used in a wide range of healthcare, food industry, and domiciliary applications and are commonly found in hard surface materials and textiles. Such extensive use raises questions about its safety, environmental toxicity, and the risks associated with microbial resistance and cross-resistance [2]. Silver nanoparticles (AgNPs) are silver minerals of very small size (10–100 nm). There are many routes to producing nanoparticles, including chemical and physical methods, but these methods have many problems, i.e., using toxic chemicals and consuming high energy. Therefore, an eco-friendly biological path has emerged [3].

Many studies have reported using plants, algae, and bacteria in the biological synthesis of nanoparticles [4,5,6]. Therefore, the green synthesis of AgNPs ensures their safe use in many biomedical applications with no side effects [7]. Additionally, they exhibited antioxidant, antimicrobial, antiviral, and antitumor activities [8]. This study used phycocyanin to synthesize AgNPs because it is rich in bioactive groups responsible for biotransformation [9]. The produced AgNPs are safer than chemotherapy; they have attracted the attention of workers in different fields all over the world due to their unique chemical and physical properties, and they have promising applications in medicine, agriculture, environmental remediation, food technology, water treatment, and multidrug resistance microbes [10,11].

Antimicrobial resistance issues have risen dramatically recently, posing a severe concern to humans worldwide. Novel compounds for pharmaceutical applications prompts are necessary to face this issue [12], where Habeeb Rahuman, et al. [13] exposed several medicinal plants used to fabricate ecofriendly AgNPs with promising biomedical applications. Furthermore, Maheshwaran et al. [14] prepared face-centered cubic Ag_2_O-NPs using the flower extracts of Zephyranthes Rosea to treat several medical conditions, such as diabetes, tuberculosis, and breast cancer, among others. The presence of Amaryllidaceae alkaloids makes the plant extract a potential agent for the biosynthesis of nanoparticles. The resulting nanoparticles were allowed to precipitate, annealed at 80 °C for 2 h, and then subjected to antibacterial, antioxidant, anti-inflammatory, and anti-diabetic assays.

Additionally, *Ficus benghalensis* is commonly used in many medicinal applications and has long been used in relieving toothaches. The authors prepared Ag_2_O-NPs by mixing the prop root extracts with the AgNO_3_ solution allowing the mixture to react under constant stirring in ambient conditions. The resulting nanoparticles were investigated for bactericidal activity against dental pathogens. In another study by Manikandan et al. [15], Ag_2_O-NPs were also successfully fabricated using the rind extracts of *Artocarpus heterophyllus*; the resulting nanoparticles were tested for antifungal activity against plant pathogenic fungi. A similar method was employed by El-Ghmari et al. [16] to utilize the aqueous extracts of *Herniaria hirsute* to synthesize Ag_2_O-NPs, which were used in the photocatalytic degradation of methylene blue dye.

However, the huge release of nanoparticles (NPs) into the environment (air, water, and soil) by several sectors is generating nano waste, posing a threat to ecosystem balance and endangering the health of living beings and having negative effects on human and animal health. NPs have increased the risk of several human disorders, including diabetes, cancer, bronchial asthma, allergies, and inflammation.

Cancer is the second highest cause of mortality after cardiovascular diseases [17]. Nanotechnology is used to develop anticancer agents to present a new strategy for combating cancer. As per Quan et al. [18], nanoparticles are active candidates for direct drug delivery to a particular location in target cancer cells with minimal adverse effects. Hung et al. [19] also stated that nanoparticles might enter cells via non-specific absorption and cell activities such as adhesion, cytoskeleton organization, migration, proliferation, and death. The form and size of nanoparticles influence these processes. According to Yakop et al. [20], the size and form of metallic nanoparticles are critical because they allow for maximum accumulation in malignant tumors. Other variables influence the pharmacokinetics and pharmacodynamics of nanoparticles [21]. According to Jeyaraj et al. [22], the cytotoxicity of Hela cell lines increased with increasing AgNPs concentrations. Mfouotunga et al. [23] also showed that the higher cytotoxicity of AgNPs on MCF7 cells was owing to reduced viability, increased cytotoxicity, and proliferation, all of which culminated in apoptosis via programmed cell death.

Therefore, in this study, we investigate the anticancer activity of green AgNPs synthesized by *Spirulina platensis* phycocyanin on two cancer cell lines, the lung cancer cell line (A-549) and breast cancer cell line (MCF-7), compared to the normal human lung cell line (A138). Additionally, we investigate the bactericidal activity against *Bacillus cereus* ATCC11778, *Staphylococcus aureus* ATCC29737, *Escherichia coli* ATCC8379, and *Klebsiella pneumonia*, as well as the fungicidal activity against *Candida albicans* (ATCC6019) and *Aspergillus niger*, and their potential use in medicinal applications.

## 2. Material and Methods

### 2.1. Phycocyanin Isolation and Purification

Phycocyanin (C-PC) was extracted from the blue-green alga, *Spirulina platensis*, according to Boussiba and Richmond [24]. Two grams of experimental algae were stirred in 200 mL of a phosphate buffer (0.1 M) pH 7.2 containing 100 µg/mL lysozyme and 10 mM EDTA in a shaking water bath at 30 °C for 24 h to test the enzymatic breakdown of the cell wall. The cell remnants were eliminated, and the slurry was centrifuged for one hour at 10,000 rpm, providing a bright blue supernatant of C-PC. C-PC crude extracts were centrifuged at 10,000 rpm for 30 min without being cooled. The supernatant containing the C-PC solution was precipitated twice by ammonium sulfate precipitation at two levels (50% and 75% (NH_4_)_2_SO_4_ (*w*/*v*) at pH 7.2 for 6 h).

Ten milliliters of ammonium sulfate extract were dialyzed against the extraction buffer using a Dialyse membrane-70. The sample was dialyzed twice against one liter of extraction buffer, first at room temperature and then overnight at 4 °C. The extracted solution was recovered from the dialyzed membrane and filtered through a 0.45 m filter.

The phycocyanin was purified using anion exchange chromatography using a DEAE-Cellulose column (30 × 2 cm) equilibrated with 150 mL of acetate buffer (pH, 5.10). Ten milliliters of dialyzed, filtered material were deposited on the column. The column was developed using a linear gradient of acetate buffer with a pH range of 3.76 to 5.10; the eluate was collected in 5 mL fractions, and the buffer flow rate was set to 20 mL h^−1^. A Specord 200 spectrophotometer (Analytik, Jena GmbH, Jena, Germany) was used to scan the sample in the 300–750 nm range for absorbance assessment. The purified phycocyanin was identified by HPLC [25].

### 2.2. Preparation of Phycocyanin

Five grams of dried phycocyanin powder were mixed with 100 mL of distilled water and incubated overnight in a rotary evaporator incubator at 30 °C and 150 rpm. The samples were filtered using Whatman’s No. 1 filter paper and kept at 4 °C for further use.

### 2.3. Biosynthesis of Spirulina Platensis Phycocyanin Silver Nanoparticles (SPAgNPs)

Approximately 0.17 g of AgNO_3_ was dissolved in 1 L of sterilized deionized water to obtain the AgNO_3_ solution (1 mM), then 10 mL of aqueous phycocyanin filtrate was added to 90 mL of AgNO_3_ and placed in optimized conditions of pH 5, a temperature of 30 °C, a reaction time of 5 h, and an agitation speed of 150 rpm [26] until the color changed to a reddish brown.

### 2.4. Characterization of Spirulina Platensis phycocyanin Silver Nanoparticles (SPAgNPs)

The SPAgNPs were characterized using six advanced devices. The UV absorbance of SPAgNPs in the 200–700 nm range was characterized using a LaxcoTM dual-beam spectrophotometer [3]. The JEOL 1010 TEM (JEOL, Tokyo, Japan) was used to assess the size of SPAgNPs [6]. The active groups in SPAgNPs involved in biotransformation and stability were identified using FTIR analysis. FT-IR spectroscopy (“Bruker Tensor 37, Kaller”, Germany) was used to detect the active compounds in SPAgNPs in the 4000 cm^−1^ to 500 cm^−1^ range using KBr pellets [27]. The accurate size and charge were assessed by DLS analysis. A zeta sizer (Nano “Z2 Malvern, UK) estimated the size of SPAgNPs, and the surface charge of SPAgNPs was measured via zeta potential [26,28]. The elemental composition of AgNPs was determined using EDX, and the nature of SPAgNPs was detected by XRD [29].

### 2.5. Biological Activity of Spirulina Platensis Phycocyanin and SPAgNPs

#### 2.5.1. Antioxidant

Saad et al. [26] examined the DPPH scavenging activity of phycocyanin and AgNPs at a concentration of (50 µg/mL) with some changes. First, 100 µL of SPAgNPs and phycocyanin were homogenized in 100 µL of DPPH for 10 min; the mixed samples were placed in wells of a microtiter plate and kept for 30 min in the dark; the ready plate was read at 517 nm using the microtiter plate reader (BioTek Elx808, Santa Clara, CA, USA) and calculated in the equation
$$\%\;Antioxidant\;activity=\frac{Control\;absorbance-Sample\;absorbance}{Control\;absorbance}\times 100$$

#### 2.5.2. Cytotoxicity Effects

##### Preparation of Different Concentrations of SPAgNPs and Phycocyanin

From a stock solution of both SPAgNPs and phycocyanin of 50 ppm, the following concentrations were prepared (50, 25, 12.5, 6.25, 3.125, 1.5652, 0.78, 0.39, 0.195, 0.097, 0.049, and 0.0244 µg/mL) via dilution with saline phosphate buffer pH 7.2.

##### Cell Culture and Maintenance

Cell lines, breast cancer cell line (MCF-7), human lung cancer cell line (A-549), and normal lung cell line (A138) were obtained from the American Type Culture Collection (ACC, Rockville, MD, USA), cultured in Engle’s medium (DMEM) accompanied by necessary growth factors. The subculture was performed after reaching 95% confluence. The growth media were renewed every three days.

##### Measurement of Cytotoxicity Test (MTT Assay)

A comparative study on AgNPs and phycocyanin’s nanotoxicity was evaluated on cancer cells using the MTT assay following Mosmann [30]. This assay is a sensitive, quantitative, and reliable colorimetric method that measures cell viability and proliferation. For the chemosensitivity test, exponentially growing cells were harvested, counted, and inoculated (at the appropriate concentrations in a volume of 100 μL) into 96-well microtiter plates; 8 replicates were prepared for each dose. U-bottom microplates were used for suspension-growing cells, whereas flat-bottom microplates were used for plastic-adherent cell cultures. Immediately or 24 h after cell seeding, 10 μL of different dilutions of drugs, prepared 10× more concentrated than requested, was added to each well. The MTT assay was performed after different incubation times at 37 °C in a humidified 5% CO_2_ atmosphere. MTT (Sigma, St. Louis, MO, USA) was dissolved at a concentration of 5 mg/mL in Hank’s salt solution and filtered with a 0.45 μ filter (in order to avoid MTT aggregates). Ten microliters of the MTT solution were added to each well and the control wells without cells. Additional controls consisted of media alone with no cells, with or without the various drugs. After 4–6 h of incubation, microtiter plates were centrifuged at 2000 rpm for 10 min; the medium was removed, and 100 µL of DMSO was added to each well. After thorough mixing with a mechanical plate mixer, the absorbance of the wells was read in a scanning well microculture plate reader at the test and reference wavelengths of 550 and 620 nm, respectively, which are approximately the peak and lowest MTT wavelengths of absorption required to avoid quenching from the growth medium, in particular, phenol red. Absorbance values from all wells were corrected against these control absorbance levels. The ID50 was defined as the concentration of drug that produced a 50% reduction in absorbance compared with untreated control cells [31].

#### 2.5.3. Antimicrobial

The antimicrobial activity of SPAgNPs and phycocyanin was determined against *Bacillus subtillis* (HTCC441), *Staphylococcus aureus* (ATCC29737), *Escherichia coli* (ATCC8379), and *Klebsiella pneumonia* (ATCC00607). The activity of SPAgNPs was also tested against unicellular and filamentous fungi, *Candida albicans* (ATCC6019), and *Aspergillus niger* (ATCC16404). These strains were selected based on the microbial examination of patients assaulted by microbial infection. It was found during microbial examination with a light microscope and biochemical and morphological tests that these isolates are the most isolates that cause critical infections in humans. These isolates were confirmed by molecular identification through isolating DNA and using PCR to detect genes. The bacterial isolates were identified based on 16S rRNA gene sequence analysis. Sequencing was performed via the automated DNA sequencer (ABI Prism 3130 Genetic Analyzer by Applied Biosystems Hitachi, Tokyo, Japan). Genomic DNA was obtained by the hexadecyltrimethylammonium bromide (CTAB) technique, and the integrity and level of purified DNA were established by agarose gel electrophoresis. The DNA level was customized to 20 ng/µL for PCR amplification. The forward primer used with the isolates is (5 AGA GTT TGA TCC TGG CTC AG 3), and the reverse is (5 GGT TAC CTT GTT ACG ACT T 3). PCR products were isolated by electrophoresis on 1.5% agarose gels stained with ethidium bromide and documented in the Alphaimager TM1200 documentation and analysis system. The obtained polymorphic differences were analyzed via the program NTSYS-PC2 by assessing the distance of isolates by Jaccard’s Similarity Coefficient. The antimicrobial activity was tested separately using the well diffusion method [5,32]. The Muller Hinton agar medium was poured into Petri-dishes and inoculated with 1 mL of microbial strain culture individually (14 × 10^6^ CFU/mL). A Cork borer was used to make a 7 mm well in MHA plates. Each well received 100 µL of phycocyanin and SPAgNPs solutions. Inoculated plates were incubated at 37 °C for 24 h. All experiments were carried out in triplicate, and the inhibitory activity was expressed as an inhibition zone diameter mean [32,33].

##### Test for Antibiotic Susceptibility for Antibiotics against Pathogenic Microbes

The guidelines for antibiotic susceptibility tests were followed by CLSI [34] using the disc diffusion method. Six antibiotics were selected according to their daily use in treating microbes. Three antibiotics were used to treat bacterial cultures (ciprofloxacin 5 g/mL, doxycycline 10 g/mL, and tetracycline 30 g/mL), and three antibiotics were used to treat fungi, Amphotericin B 30 µg/mL (AMP), Cefrazidine 30 µg/mL (Caz), and Liposomal amphotericin B 50 g/mL (L-AMP).

Muller Hinton agar plates and potato dextrose agar were prepared, inoculated with all negative and positive control microbial strains, and then the antibiotics discs were placed at the plate’s sides. The MHA and PDA plates were then incubated at 37 °C for 24 h and 28 °C for three days, respectively [34,35,36].

All experiments were performed in triplicate. The inhibition zone diameters of antibiotics were measured (mm). Multidrug-resistant microbes were defined as isolates that were resistant to two or more commercial antibiotics [37]. The most susceptible isolate was selected for further studies.

## 3. Results and Discussion

Algal synthesis of AgNPs is especially interesting because of the high capacity of the algae to take in metals and reduce metal ions [38]. In line with this, Bishoyi, et al. [39] fabricated silver nanoparticles using the brackish water blue-green alga *Oscillatoria princeps*. Furthermore, plants were used to produce AgNPs. On the other hand, Marchiol et al. [40] produced silver nanoparticles from extracts of multiple organs (leaves, stems, and roots) of several plants, including *Brassica juncea*, *Festuca rubra*, and *Medicago sativa*, in comparative research. After 48 h, the color of the AgNO_3_ solution changed from pale green to dark brown when phycocyanin was applied. The emergence of a dark brown color after mixing the solutions indicated AgNO_3_ biotransformation into AgNPs. The highest AgNPs yield was obtained by adding phycocyanin to the AgNO_3_ solution at a ratio of 1:90; *v*:*v*. Our research found no precipitation in SPAgNPs. The absence of precipitation in the transformation of silver nanoparticles shows that the nanoparticles were created with tiny particle sizes.

A variety of nanoparticles were created using phenolic extracts [4]. Plant organ extracts behaved differently throughout the nanoparticle synthesis process [4]. In this study, the enhanced total phenolic content in phycocyanin improves the reduction of Ag^+^ to nanoscale-sized silver particles, which shortens the manufacturing process of AgNPs by polyphenols in phycocyanin.

### 3.1. Characterization of AgNPs

The UV-Vis spectrum revealed that the phycocyanin formed varying amounts of silver nanoparticles. After 72 h, the UV absorption spectra in the 200–700 nm range were examined to confirm silver nanoparticle production and stability. At 400 nm, AgNPs had the most significant absorbance of 0.6 a.u. (absorbance unit) (Figure 1A). The physical and chemical features of the SPAgNPs generated in this work are consistent with the findings of Pirtarighat et al. [41], who discovered that by using *Salvia Spinosa*, AgNPs displayed SPR at 450 nm, indicating the synthesis of AgNPs.

The morphological characteristics of SPAgNPs are depicted in Figure 1B. They were examined using TEM. The size of spherical SPAgNPs was in the range of 9–55 nm. The variance in size was determined mainly by the phenolic component concentration of the extracts used to lower Ag+, as well as other optimization conditions. Padalia et al. [29] also created *Calendula officinalis-* AgNPs with diameters ranging from 10 to 90 nm and shapes of spherical to hexagonal to irregular forms.

In Figure 1C, and D, the DLS findings of SPAgNPs reveal a single peak. The exact size was 30 nm, with a −26.32-mV net negative charge. The negative surface charge on nanoparticles supports their stability. In this regard, Kratoová et al., [42] discovered that sample values between +30 mV and 30 mV were stable and did not aggregate.

The EDX image (Figure 1E) reveals that silver content was predominant. These typical optical peaks at ~3 KeV were attributed to the surface plasmon resonance (SPR) of metallic Ag nanocrystals.

The exact nature of the formed SPAgNPs may be deduced from the XRD spectrum of the sample. The XRD spectrum (Figure 1F) in this study using SPE confirmed that the silver particles formed were in the form of nanocrystals, as evidenced by the peaks at 2θ values of 33.12°, 46.16°, 57.19°, and 83.17° corresponding to (111), (200), (220), and (311) planes, respectively, for silver. However, the intensity of peaks for plane (111) was stronger. Few other obvious peaks at different 2θ angles such as 27, 37, 65, and 75° can be attributed to silver chloride and other ions used in the culture medium preparation, indicating that the biosynthesized silver nanoparticles are well crystallized. Manivasagan et al. [43] reported that these typical XRD peaks occur due to the presence of the Face-centered cubic (Fcc) of the crystalline silver nanoparticles. The high-intensity peak for FCC materials is generally (111) reflection, which is observed in all samples. The XRD shows that SPAgNPs are crystalline to 30 nm.

Figure 1G,H illustrate the FTIR spectrum of SPAgNPs and SPE. Eleven bands were detected in SPE, while only eight bands were observed in FTIR of SPAgNPs, indicating that some active groups from SPE coated the SPAgNPs. According to the FTIR spectra, the tested SPAgNPs were surrounded by a reducing agent, phycocyanin. Eight bands were ascribed to active groups in the SPAgNPs suspension (3455, 2901, 1617, 1599, 1130, 950, 721, and 440 cm^−1^). The NH groups occurred at 3455 and 1617 cm^−1^, respectively.

Furthermore, aliphatic CH occurred at approximately 2901 cm^−1^. In addition, the COO appeared at 1130 cm^−1^. These findings suggested the existence of a protein-phenolic complex and protein-coated AgNPs, which might help to reduce and stabilize biosynthesized SPAgNPs. Our investigation’s active groups in SPAgNPs are consistent with El-Bendary et al. and [44]. El-Saadony et al. [2].

### 3.2. Biological Activities of Phycocyanin and SPAgNPs

The anticancer and antimicrobial activity and mechanism of SPAgNPs are displayed in Figure 1.

#### 3.2.1. Antioxidant Activity

Phycocyanin (50 ppm) scavenged 65% of DPPH. DPPH scavenging activity was boosted by 30% in SPAgNPs (50 ppm) compared to ascorbic acid, which scavenged 90% of DPPH˙ (Figure 2). Because of their propensity to donate electrons, phenolic compounds are potent antioxidants with a high reducing capacity [45]. Saad et al. [26] discovered that AgNPs manufactured using pomegranate and watermelon peel extracts scavenged 85% of DPPH, a 25% increase compared to the pomegranate and watermelon peel extracts.

#### 3.2.2. Cytotoxicity Effect of Phycocyanin and SPAgNPs

In the present study, the MTT assay was conducted to evaluate the cytotoxicity of AgNPs and phycocyanin extract against two cell lines, MCF 7 (breast cancer cell line) and A549 (Human lung carcinoma), compared to A138 (Normal human lung cell line) [30]. The MTT test depends on the capacity of mitochondrial lactate dehydrogenase (LDH) in live cells to transform the water-soluble substrate, MTT, into water-insoluble dark blue formazan. Dimethyl sulfoxide is used as a solubilizer to transform the insoluble purple formazan result into a colored solution. Its absorbance was measured spectrophotometrically at a wavelength of 500–600 nm, according to Cory et al. [46].

Twelve concentrations of phycocyanin and SPAgNPs were prepared and monitored for their effects on the viability and proliferation of the cancer cell lines regarding their metabolic reduction potency. It was found that SPAgNPs exhibited a cytotoxic effect against both normal and cancer cell lines in a dose-dependent manner. The dramatic effect of SPAgNPs on the viability of treated cell lines shown in Table 1, Table 2 and Table 3 clarified that as the concentrations of SPAgNPs increased, a significant reduction in cell viability was observed. On the other hand, the highest percentage of cell viability was recorded at the lowest concentration of SPAgNPs (0.0244 µg/mL) since they were 100, 141.17, and 81.28%.

However, lower cell viability was recorded at the highest concentration of SPAgNPs and phycocyanin (50 ppm), where they were 10.9, 9.01, and 25.77% against MCF 7, A549, and A138, respectively. Moreover, according to IC 50, the three cell lines can be arranged ascending as A138, A549, and MCF 7, where IC50 of SPAgNPs was 0.32, 12.96, and 26.55 µg/mL, while in phycocyanin, it was 1.77, 20.46, and 32.16 µg/mL, respectively (Table 1, Table 2 and Table 3 and Figure 3). It could be stated that both SPAgNPs and the phycocyanin extract showed a dose-dependent response in all tested cell lines. The inhibitory percentage also confirmed the results. Finally, SPAgNPs showed excellent anticancer activity against all tested cell lines compared with phycocyanin extract. These results agree with Nayak et al. [47], who confirmed a significant antiproliferative activity of biosynthesized AgNPs; such activity may be due to the synergistic effect of both nano-sized silver and the bioactive compounds attached to the surface of nanoparticles. These results parallel those of Moldovan et al. [48].

Donga and Chanda [49] found that the cytotoxicity of SPAgNPs on Hela cell lines increased with SPAgNPs’ concentration dependence. Furthermore, Barabadi, et al. [50] stated the cytotoxicity of AgNPs against cervical cancer cells. The mechanism of SPAgNPs toxicity toward cancer cell lines could be explained by Mfouotyngo et al. [23] who stated that the enhanced cytotoxic activity of SPAgNPs on MCF7 cells was due to the increased cytotoxicity, decreased cell viability, and proliferation, which causes apoptosis through induced programmed cell death. Furthermore, Nikzamer et al. [51] confirmed that this cytotoxicity might be caused by the active physicochemical interaction of silver atoms with the functional groups of proteins, nitrogen bases, and phosphate groups in DNA. The increased production of ROS (reactive oxygen species) leads to the activation of pro-apoptotic proteins, which initiates DNA damage in the form of DNA fragmentation, leading to apoptotic cell death.

#### 3.2.3. Antimicrobial Activity

##### Antibacterial

SPAgNPs and phycocyanin were tested to detect their antimicrobial activity against studied pathogenic bacteria using the well diffusion assay, and the diameter of the inhibition zones was measured (mm).

Measurement of inhibition zones around agar wells demonstrated the dose-dependent activity of SPAgNPs against selected clinical pathogens such as *Staphylococcus aureus*, *Bacillus cereus*, *Escherichia coli*, and *Klebsiella pneumonia*. Results in Table 4 and Table 5 clarified that the agar well diffusion assay showed no inhibition zones in bacterial strains exposed to low concentrations of SPAgNPs (5, 10, and 20 µg/mL); however, the inhibition zones of 3.2, 2.06, 2.00, and 2.03 mm occur at higher concentrations (30, 40, and 50 µg/mL). At the same time, phycocyanin (40 µg/mL) achieved inhibition zones (1.4, 1.17, 1.166, and 0.67 mm) against *Staphylococcus aureus*, *Bacillus cereus*, *Escherihia coli*, and *Klebsiella pneumonia,* respectively. Table 4 and Table 5 show that higher concentrations of SPAgNPs (30, 40, and 50 µg/mL) showed higher inhibition zones than positive controls (standard antibiotics), particularly against the Gram-positive bacteria *Staphylococcus aureus* and *Bacillus cereus*. Moreover, SPAgNPs (50 µg/mL) has a similar inhibitory effect on doxycycline since the estimated clear zones were 2.5 and 2.57 mm, respectively. The results showed that the vulnerability of Gram-negative bacteria to SPAgNPs and phycocyanin was lower than that of Gram-positive bacteria because of the presence of thick cell walls [52].

The minimum inhibitory concentration (MIC) of SPAgNPs and phycocyanin was 30 and 40 µg/mL against Gram-positive and Gram-positive bacteria, respectively. Therefore, AgNPs are stronger bactericides than phycocyanin. These results agree with Hamad [53], who biosynthesized AgNPs with an excellent antibacterial effect on Gram-positive and Gram-negative bacterial strains. Crisan et al. [54] found that AgNPs fabricated using aqueous leaf extract of *Corchorus capsularis* exhibited considerable antimicrobial activity against multidrug-resistant (MDR) *P. aeruginosa*, *Staphylococcus aureus,* and CoNS isolates from post-surgical wound infections. Additionally, AgNPs synthesized from the *Bacillus subtilis* efficiently revealed antibacterial activity of nanoparticles against pathogens and can be used for antimicrobial studies and may be applied in the field of nanomedicine [55].

Furthermore, Pannerselvam et al. [56] tested the antibacterial activity of AgNPs against three wound-infecting Gram-positive and Gram-negative pathogenic bacteria viz., *B. subtilis, S. aureus, M. luteus, E. coli, K. pneumoniae,* and *P. aeruginosa* at 10 µg concentrations. *P. aeruginosa* was found to be remarkably sensitive to the AgNPs with an inhibition zone of 20.3 ± 1.86 mm. The standard antibiotics of streptomycin, gentamycin, ampicillin, and erythromycin at 10 µg, when tested against the bacteria, revealed that gentamycin showed high antibacterial activity against all six wound-infecting pathogenic bacteria. The MIC and MBC concentrations evaluated against the bacteria were 4.0 ± 1.00 µg/mL and 6.3 ± 0.47 µg/mL for *S. aureus*, respectively.

Maximal antibacterial activity was due to smaller particles of AgNPs and a large surface area, which consequently penetrated the cell wall, leading to the death of bacterial cells. Kumarasamyraja and Jeganathan [57] showed similar observations. They reported that the damage caused to bacterial membranes by AgNPs depends on their size and shape, which induces the release of reactive oxygen species (ROS), forming free radicals with decisive bactericidal action.

##### Antifungal Activity

The antifungal potential of SPAgNPs and phycocyanin against unicellular fungus (*Candida albicans* ATCC 60193) and filamentous fungus (*Aspergillus niger* ATCC 16404) was estimated using an agar well diffusion assay. Table 6 and Table 7 clarify that the low concentrations of SPAgNPs and phycocyanin do not affect fungal growth. The fungicidal activities of SPAgNPs and phycocyanin were recorded at higher concentrations. The inhibition zones of SPAgNPs at levels of 30, 40, and 50 ppm were recorded as 2.33, 2.47, and 2.77 mm against the unicellular fungus *Candida albicans*, and 2.87, 3.1, and 3.47 mm in the multicellular fungus *Aspergillus niger*, respectively.

On the other hand, the fungicidal effect of phycocyanin shown in Table 7 clarifies that the recorded zone of inhibition was less than those of SPAgNPs; the inhibition zones were 1.90 and 2.27 mm at concentrations of 40 and 50 ppm against *Candida albicans* and 2.47 and 2.73 mm at the same concentrations in *Aspergillus niger,* respectively. The fungicidal effect of SPAgNPs and phycocyanin against *Aspergillus niger* was more inhibitory than their positive controls. However, the fungicidal effect of SPAgNPs and phycocyanin on *Candida albicans* was less than their values in the corresponding positive control (Table 6 and Table 7).

These results agree with Kathiravan et al. [58], stating that AgNPs have a more substantial effect on filamentous fungi than unicellular ones. AgNPs were also very effective against plant phytopathogenic fungi.

The AgNPs fabricated using *Kleinia grandiflora* leaf extract at the concentration of 200 µg/mL showed considerable antimicrobial activity against *Pseudomonas aeruginosa* and *Aspergillus niger* [59]. Furthermore, Mansoor, et al. [60] reported that AgNPs inhibited the growth of various fungi, including *Aspergillus fumigates, Aspergillus niger, Aspergillus flavus, Trichophyton rubrum, Candida albicans,* and *Penicillium* species.

According to Kim et al. [61], the electrostatic attraction between the negatively charged cell membranes of microorganisms, such as bacteria, viruses, and fungus, and the positively charged nanoparticles is critical for these particles’ antibacterial activity. Furthermore, AgNPs generate ROS and free radicals, damaging cell walls and components. As a result, they modify the permeability of the cell membrane, resulting in cell death [62].

However, the advantages of nanoparticles are evident in several industries, including electronics, agriculture, chemicals, medicines, and food. Metal oxide NPs, such as ZnO, SiO_2_, CeO, TiO_2_, Al (OH)_3_, CuO, Ag, nanoclays, carbon nanotubes, and nano-cellulose, are the most frequently used chemical NPs across a variety of industries. Nonetheless, the huge release of nanoparticles (NPs) into the environment (air, water, and soil) by several sectors is generating nano waste, posing a threat to ecosystem balance and endangering the health of living beings. Size, nature, reactivity, mobility, stability, surface chemistry, aggregation, and storage duration are only a few of the features that influence the toxicity of nanoparticles. NPs have negative effects on human and animal health. NPs have increased the risk of several human disorders, including diabetes, cancer, bronchial asthma, allergies, and inflammation. The reproductive systems of animals have also been influenced by the toxicity of NPs such as Au, TiO_2_, etc. NPs are absorbed by cells via phagocytosis and endocytosis after entering the body of an animal by ingestion and inhalation. They produce reactive oxygen species (ROS), which cause lipid peroxidation and mitochondrial damage. As a result of several NPs, including Ag, Cu, ZnO, and Ni, diverse bacteria enzyme activity has also been diminished. In addition, excessive NP production disrupts the food web of the environment [63,64,65].

## 4. Conclusions

A biological approach for producing silver nanoparticles utilizing the proteinaceous pigment phycocyanin found in cyanobacterial cells has been disclosed. SPAgNPs are employed efficiently in medical and pharmaceutical, food, and cosmetic sectors due to their improved antibacterial, anticancer, and antihemolytic activities. *Spirulina plantlets* phycocyanin was effectively employed in the production of SPAgNPs. UV-Vis spectroscopy, zeta potential, FTIR, TEM, and XRD characterization indicated the production of SPAgNPs. The current work investigates SPAgNPs’ potential antibacterial effectiveness against Gram-positive and Gram-negative microorganisms. SPAgNPs showed in vitro anticancer activity against lung cancer cell lines (A-549), breast cancer cell lines (MCF-7), and human regular lung cell lines (A138).

## Data Availability

The data presented in this study are available on request from the corresponding authors.

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
