# Peer review of "Evaluation of Green Silver Nanoparticles Fabricated by Spirulina platensis Phycocyanin as Anticancer and Antimicrobial Agents"

_life, 2022, doi:10.3390/life12101493_

Round 1

Reviewer 1 Report

Line 28: scav-enged

Line 35: One of the cell lines is your control (A138), so you tested on only two cancer cell lines not three.

Line 173 Equation is misspelt

Line 196: “Read at”

The authors mentioned that green NPs are effective against MCF-7 and A-549. Figure-3 shows individual cell line viability % compared among Spirulina, AgNPs and DOX. However, the toxicity of NPs is high on normal lung cancer cells (A138). Plotting a graph against all three cell line models reveals this purpose. I recommend to review your design and experimental approach to address my question.

Author Response

Reviewer 1 comments

Comments and Suggestions for Authors

Line 28: scav-enged

Response: Done accordingly

Line 35: One of the cell lines is your control (A138), so you tested on only two cancer cell lines not three.

Response: it was adjusted accordingly

Line 173 Equation is misspelt

Response: It was revised accordingly

Line 196: “Read at”

Response: Done accordingly

The authors mentioned that green NPs are effective against MCF-7 and A-549. Figure-3 shows individual cell line viability % compared among Spirulina, AgNPs and DOX. However, the toxicity of NPs is high on normal lung cancer cells (A138). Plotting a graph against all three cell line models reveals this purpose. I recommend to review your design and experimental approach to address my question.

Response: In Figure 3, we only present the effects of SPE, AgNPs, and DOX against the carcinoma cell lines MCF-7 and A-549. Based on the results in Table 3, the additives, SPE, AgNPs, and DOX at 50 ppm affected the viability of A138 cells. However, they did not have the same effect on the carcinoma cells, MCF-7 and A-549, where there was an increase in the viability of A-138 cells by up to 150–160% compared to the MCF-7 and A-598 cancer cells, indicating that AgNPs affect both cancer and normal cells, but the effect on carcinoma is higher than normal cells, with less toxicity compare

Reviewer 2 Report

The manuscript submitted to journal “LIFE”, titled “Characterization and Evaluation of Green Silver Nanoparticles Fabricated by Spirulina Platensis Phycocyanin as Anticancer agents Against Mcf-7 and A-549 Cancer Cell Lines and Antimicrobial agents Against Pathogenic microorganisms” demonstrates the green synthesis of silver nanoparticles using Spirulina platensis phycocyanin extract. The as-prepared nanoparticles were tested for their various biological properties including anticancer, antimicrobial and antifungal properties. The objectives of the research are well-founded as green synthesis of nanomaterial provides a suitable alternative for the preparation of biocompatible nanoparticles and is also consider crucial for other environmental benefits. However, in the submitted manuscript there are several drawbacks which, in the opinion of the reviewer, withhold the positive acceptance for publication in the current state. For instance, the manuscript is poorly written, there is no coordination among different paragraphs of the introduction. Too many references have been discussed both in the introduction and in results and discussion in the form of review which is unnecessary. Instead the authors should have focused more on the importance of green synthesis in the perspective of biomedical applications.

There are several grammatical mistakes in the manuscript. The experimental methods are poorly explained.

Furthermore, the level of characterization of the materials is insufficient to provide a clear indication of the formation of Ag NPs except the UV-Analysis TEM and EDX.  Most important characterization methods such as XRD is missing, although the authors have written in the conclusion that they have measured, but the diffractogram is missing in the manuscript.

Authors have indicated that smaller size silver nanoparticles were formed, however they mentioned that the size 25 to 80 nm, which is too big. Moreover, the TEM image given in Figure 1 does not clearly reflect that. There is huge discrepancy between the TEM image and size distribution figure. The authors should provide a clear TEM image and recalculate the size distribution of the particles.

Provide high resolution image of FT-IR and compare the IR spectrum of Ag NPs with that of pure extract.

Instead of putting all the important figures in one image, authors should distribute the figures in different images with clear caption.

Author Response

Reviewer 2 comments

Comments and Suggestions for Authors

 The manuscript submitted to journal “LIFE”, titled “Characterization and Evaluation of Green Silver Nanoparticles Fabricated by Spirulina Platensis Phycocyanin as Anticancer agents Against Mcf-7 and A-549 Cancer Cell Lines and Antimicrobial agents Against Pathogenic microorganisms” demonstrates the green synthesis of silver nanoparticles using Spirulina platensis phycocyanin extract. The as-prepared nanoparticles were tested for their various biological properties including anticancer, antimicrobial and antifungal properties. The objectives of the research are well-founded as green synthesis of nanomaterial provides a suitable alternative for the preparation of biocompatible nanoparticles and is also consider crucial for other environmental benefits. However, in the submitted manuscript there are several drawbacks which, in the opinion of the reviewer, withhold the positive acceptance for publication in the current state. For instance, the manuscript is poorly written, there is no coordination among different paragraphs of the introduction. Too many references have been discussed both in the introduction and in results and discussion in the form of review which is unnecessary. Instead the authors should have focused more on the importance of green synthesis in the perspective of biomedical applications.

Response: Thanks for the reviewer for his valuable suggestions and comments that greatly enhanced the manuscript. The introduction was rearranged, the extra citations were deleted, and new citations were added related to the importance of green synthesis route in medical applications

There are several grammatical mistakes in the manuscript. The experimental methods are poorly explained.

Response: The experimental were more detalied

Furthermore, the level of characterization of the materials is insufficient to provide a clear indication of the formation of Ag NPs except the UV-Analysis TEM and EDX.  Most important characterization methods such as XRD is missing, although the authors have written in the conclusion that they have measured, but the diffractogram is missing in the manuscript.

Response: XRD image was added

Authors have indicated that smaller size silver nanoparticles were formed, however they mentioned that the size 25 to 80 nm, which is too big. Moreover, the TEM image given in Figure 1 does not clearly reflect that. There is huge discrepancy between the TEM image and size distribution figure. The authors should provide a clear TEM image and recalculate the size distribution of the particles.

Response: Thanks for the reviewer for this comment. Clear TEM image was provided and the average size of TEM was carefully recalculated and correlated with size distribution

Provide high resolution image of FT-IR and compare the IR spectrum of Ag NPs with that of pure extract.

Response: high resolution FT-IR images of SPAgNPs and pure extract were provided

Instead of putting all the important figures in one image, authors should distribute the figures in different images with clear caption.

Response: Thanks for the reviewer for his valuable suggestions and comments, the important images were detailed

Reviewer 3 Report

The manuscript entitled " Characterization and Evaluation of Green Silver Nanoparticles 3 Fabricated by Spirulina Platensis Phycocyanin as Anticancer agents Against Mcf-7 and A-549 Cancer Cell Lines and Antimicrobial agents Against Pathogenic microorganisms

" was reviewed carefully. I found this manuscript attractive for the researchers in the field of nanomedicine for biomedical application. The quality of English needed improvements. Furthermore this manuscript needed the following essential revisions. The following comments will help to increase the quality of the manuscript before publication.

1. Title of the manuscript is not appropriate and Make it clear and very precise title

2. When i studied the section of "Introduction", i was not satisfied from the structures and information. Introduction part authors should emphasis on Anticancer Nanomedicine and drug resistance microorganism and their impact on the development of new therapeutic approach. The following reference appropriate for citation to enhance the quality and reader visibility

https://doi.org/10.1007/s12010-022-04072-7

https://doi.org/10.1049/nbt2.12078

3. Section 2 .1 Materials and Methods authors should mentioned how to identified and purified Phycocyanin From blue-green alga, Spirulina platensis,

4. In the section 3.1 for the characterization data SEM image author mentioned size of AgNPs with diameters ranging from 10 to 90 nm and shape of spherical to hexagonal to irregular forms. I am not got the same shape in the TEM Image (b). Authors should provide the high resolution of Sem images should taken again replace with old image.  All imeges Fig 1 should be given in High resolution.

5. Section 3.3.and Cytotoxicity effect of phycocyanin and AgNPs

The result and discussion part author should discuss their result with  recent relevant AGNPS references with Anticancer activity 

https://doi.org/10.1016/j.micpath.2017.04.013

Ihttps://doi.org/10.1007/s10876-019-01697-3

·       https://doi.org/10.1080/24701556.2019.1583251

6. Section 3.4 Antimicrobial activity needed improvement for the result and discussion section compare with appropriate recent relevant references. Authors should remove irrelevant references in this section.

https://doi.org/10.1007/s10876-019-01583-y

https://doi.org/10.1166/jbns.2014.1201

https://doi.org/10.1007/s10876-020-01759-x

7. The possible disadvantages of metallic based nanoparticles for human should be mentioned in the section of "Introduction" and "Discussion".

8. The authors are suggested a schematic figure describing the proposed Anticancer and antibacterial activity of the inorganic nanoparticles.

Author Response

Reviewer 3 comments

Comments and Suggestions for Authors

The manuscript entitled " Characterization and Evaluation of Green Silver Nanoparticles 3 Fabricated by Spirulina Platensis Phycocyanin as Anticancer agents Against Mcf-7 and A-549 Cancer Cell Lines and Antimicrobial agents Against Pathogenic microorganisms " was reviewed carefully. I found this manuscript attractive for the researchers in the field of nanomedicine for biomedical application. The quality of English needed improvements. Furthermore, this manuscript needed the following essential revisions. The following comments will help to increase the quality of the manuscript before publication.

Response: Thanks for the reviewers for the positive comment on our manuscript. The language was revised by an expert, and all the following comments and suggestions were considered

Title of the manuscript is not appropriate and Make it clear and very precise title

Response: The title was enhanced

  1. When i studied the section of "Introduction", i was not satisfied from the structures and information. Introduction part authors should emphasis on Anticancer Nanomedicine and drug resistance microorganism and their impact on the development of new therapeutic approach. The following reference appropriate for citation to enhance the quality and reader visibility

https://doi.org/10.1007/s12010-022-04072-7, https://doi.org/10.1049/nbt2.12078

Response: The relevant references were considered and added

  1. Section 2 .1 Materials and Methods authors should mentioned how to identified and purified Phycocyanin From blue-green alga, Spirulina platensis,

Response: It was detailed accordingly, also, all methods were detailed

  1. In the section 3.1 for the characterization data SEM image author mentioned size of AgNPs with diameters ranging from 10 to 90 nm and shape of spherical to hexagonal to irregular forms. I am not got the same shape in the TEM Image (b). Authors should provide the high resolution of Sem images should taken again replace with old image.  All imeges Fig 1 should be given in High resolution.

 Response: Clear TEM image was provided, also all images were detailed in high resolution. Regarding “AgNPs with diameters ranging from 10 to 90 nm and shape of spherical to hexagonal to irregular forms” it was a result from previous study not our study. The SPAgNPs in our study is spherical

  1. Section 3.3.and Cytotoxicity effect of phycocyanin and AgNPs

The result and discussion part author should discuss their result with  recent relevant AGNPS references with Anticancer activity 

 https://doi.org/10.1016/j.micpath.2017.04.013, Ihttps://doi.org/10.1007/s10876-019-01697-3, https://doi.org/10.1080/24701556.2019.1583251

Response: The relevant references were considered and added

  1. Section 3.4 Antimicrobial activity needed improvement for the result and discussion section compare with appropriate recent relevant references. Authors should remove irrelevant references in this section.

https://doi.org/10.1007/s10876-019-01583-y, https://doi.org/10.1166/jbns.2014.1201, https://doi.org/10.1007/s10876-020-01759-x

 Response: The relevant references were considered and added

  1. The possible disadvantages of metallic based nanoparticles for human should be mentioned in the section of "Introduction" and "Discussion".

Response: it was added accordingly  

  1. The authors are suggested a schematic figure describing the proposed Anticancer and antibacterial activity of the inorganic nanoparticles.

 Response: Thanks for the reviewer for this suggestion, the scheme was added accordingly

Round 2

Reviewer 1 Report

Thank you for addressing my questions. The explanation provided is satisfactory.

Author Response

Reviewer 1

Comments and Suggestions for Authors

Thank you for addressing my questions. The explanation provided is satisfactory.

Response: We greatly thank the reviewer for his valuable efforts in reviewing and enhancing our manuscript.

Reviewer 2 Report

The authors have performed most of the suggested revision now the manuscript can be accepted in present form

Author Response

Reviewer 2

Comments and Suggestions for Authors

The authors have performed most of the suggested revision now the manuscript can be accepted in present form

Response: We greatly thank the reviewer for his valuable efforts in reviewing and enhancing our manuscript.

Reviewer 3 Report

 The revised Manuscript entitled "  Characterization and Evaluation of Green Silver Nanoparticles Fabricated by Spirulina Platensis Phycocyanin as Anticancer agents Against Mcf-7 and A-549 Cancer Cell Lines and Antimicrobial agents Against Pathogenic microorganisms". The author improved the quality of the manuscript's content by revising it in response to the reviewer's remarks and recommendations. The following minor updates to the manuscript were still required before acceptance.

1. The title of the manuscript has to be changed to something more appropriate that is short and precise based on the major conclusion of your research.

2. Check the overall check flow of the information and minor spell check required 

Author Response

Reviewer 3

Comments and Suggestions for Authors

 The revised Manuscript entitled “Characterization and Evaluation of Green Silver Nanoparticles Fabricated by Spirulina Platensis Phycocyanin as Anticancer agents Against Mcf-7 and A-549 Cancer Cell Lines and Antimicrobial agents Against Pathogenic microorganisms". The author improved the quality of the manuscript's content by revising it in response to the reviewer's remarks and recommendations. The following minor updates to the manuscript were still required before acceptance.

Response: Thanks for the reviewer. We carefully responded to all comments and suggestions.

  1. The title of the manuscript has to be changed to something more appropriate that is short and precise based on the major conclusion of your research.

Response: The title was changed accordingly.

  1. Check the overall check flow of the information and minor spell check required 

Response: The manuscript was carefully checked.